# Influence of Three Extraction Methods on the Physicochemical Properties of Kefirans Isolated from Three Types of Animal Milk

**DOI:** 10.3390/foods11081098

**Published:** 2022-04-12

**Authors:** Chiara La Torre, Alessia Fazio, Paolino Caputo, Antonio Tursi, Patrizia Formoso, Erika Cione

**Affiliations:** 1Department of Pharmacy, Health and Nutritional Sciences, Department of Excellence 2018–2022, University of Calabria, Edificio Polifunzionale, 87036 Rende, CS, Italy; latorre.chiara@libero.it (C.L.T.); patrizia.formoso@unical.it (P.F.); erika.cione@unical.it (E.C.); 2Department of Chemistry and Chemical Technologies, University of Calabria, 87036 Rende, CS, Italy; paolino.caputo@unical.it (P.C.); antonio.tursi@unical.it (A.T.)

**Keywords:** kefiran, temperature, ultrasound, extraction yield, thermal properties, microscopy morphologies

## Abstract

Kefiran is a heteropolysaccharide biopolymer usually extracted from kefir grains cultured in cow milk. Due to the lack of information on exopolysaccharides from other types of animal milk, in the present study, cow, buffalo and goat milks were used as raw materials for fermentation. The kefiran extractions from kefir grains were carried out with cold water (method I), hot water (method II) and mild heated water-ultrasound (method III), and then the recovery yield and the physicochemical properties of the kefirans were evaluated to establish the influence of both the extraction conditions and the type of milk. The highest yield was recorded for the cow kefiran using method III (4.79%). The recoveries of goat and buffalo kefirans with methods II and III were similar (2.75–2.81%). Method I had the lowest yields (0.15–0.48%). The physicochemical characteristics were studied with Fourier Transform-Infrared Spectroscopy (FT-IR), Scanning Electron Microscopy (SEM), and Differential Scanning Calorimetry (DSC). Fourier-transform infrared spectroscopy showed the same qualitative profile for all the samples, regardless of the method and the type of milk, confirming that the extraction methods did not affect the chemical structure of the kefirans. Otherwise, the thermal and morphological features of the samples showed differences according to both the type of the milk and the extraction method. The kefiran samples were very thermally stable, having a temperature of degradation (T_d_) in the range from 264 to 354 °C. The resulting morphological and thermal differences could lead to different practical applications of kefirans in the fields of nutrition and pharmacology.

## 1. Introduction

Kefiran is a postbiotic polymer produced by symbiotic microorganisms in kefir grains as the result of a fermentation process in milk. Postbiotics have health properties that help strengthen the human immune system, and for this reason, the consumption of fermented drinks increased during the COVID-19 pandemic. Recently, the most widely consumed fermented drinks worth mentioning are kombucha from tea, which exhibits more nutritional value than the starting tea [1], and the kefir beverage produced by fermenting milk with kefir grains and recognized as a probiotic dairy product [2]. Kefir grains include a cluster of symbiotic microorganisms, such as lactic acid bacteria (LAB), acetic acid bacteria, and yeast. During milk fermentation, the activity of the symbiotic microorganisms naturally leads to an increase of the vitamin content of the milk, that further rises because of the addition of prebiotics such as pectins [3]. The symbiotic microorganisms in kefir grains are homofermentative and heterofermentative lactic acid bacteria, and lactose-assimilating and non-lactose-assimilating yeasts. Furthermore, the LABs, mainly *Lactobacillus kefiranofaciens,* are known to produce extracellular polysaccharides (EPS) [4] called kefiran. In particular, *Lactobacillus kefiranofaciens* transforms lactose into lactic acid during fermentation [5,6]. Accumulated lactic acid normally inhibits the cell growth of *L. kefiranofaciens*, but it was consumed by non-lactose-assimilating yeasts that survive by consuming lactic acid as a carbon and energy source [7]. Kefiran is a glucogalactan that contains units of D-glucose and D-galactose in equal proportions [6]. The kefiran backbone is composed of a pentasaccharide unit randomly linked to one or two saccharides [8,9] by glycosidic bonds that can be fermented by colonic bacteria [10].

Kefiran is currently used in the food industry as an emulsifier [11], but is also used in the preparation of pharmaceutical products and functional foods [12]. In addition to numerous industrial applications resulting from to its good stability over a wide range of pH and temperatures and the ability to form edible films for use in food packaging [13,14], kefiran exhibits biological activities such as antibacterial, antifungal, wound healing, and antitumor properties [15,16]. The most common procedure for kefiran extraction from kefir grains involves hot water because it is an economic and simple method but, unfortunately, it could lead to low extraction yields due to the high temperatures and long times required [11,17,18]. For this reason, recently the use of ultrasound was introduced in combination with high temperatures, which allowed a reduced extraction time [19] for cow kefiran. In view of these findings, in the present study, three kefiran extraction procedures, cold and hot water, and ultrasound combined with hot water, were compared to evaluate the influence of these three methods on the physicochemical properties and the recovery of the extracted kefiran. Moreover, to the best of our knowledge, there are no data in the literature regarding the isolation of exopolysaccharides from grains cultured in buffalo and goat milks. The hypothesis of this study was that the different compositions of the various types of milk could lead to different biopolymers resulting from fermentation conducted under the same experimental conditions. To refute or confirm the hypothesis, three types of animal milk (cow, buffalo, and goat) were used for the growing of kefir grains from which to extract microbial exopolysaccharides using three methods. The biopolymers were analysed for their physicochemical characteristics to establish a preliminary identification, based on their sugar and protein composition, thermal properties, and microscopic morphologies.

## 2. Materials and Methods

### 2.1. Starter Cultures and Reagents

The milk kefir grains and plastic sieves were from Kefiralia (Arrasate, Gipuzkoa, Spain). They were composed, according to the label provided by the manufacturer, of 10^9^ CFU/g of LAB (*Lactococcus lactis* subsp. *lactis*, *Lactococcus lactis* subsp. *lactis biovar diacetylactis*, *Lactococcus lactis* subsp. *cremoris*, *Leuconostoc mesenteroides* subsp. *cremoris*, *Lactobacillus kefyr*), *Candida kefyr*, and *Saccharomyces unisporus* subsp. The milks used were purchased with the following brands: Granarolo (UHT [ultra-high temperature] cow milk), Eurospin (UHT goat milk) and fresh buffalo milk (Naples, Italy) that had been pasteurized at 90 °C for 15 min before use.

Albumin (analytical standard) and Bradford’s reagent were purchased from Thermo Fisher (Milan, Italy); sulfuric acid, phenol, buffer solutions for pH meter calibration (pH 4 and 7), and dimethyl sulfoxide (DMSO) were purchased from Sigma Aldrich (Milan, Italy).

### 2.2. Activation of Kefir Grains and Fermentation of Milk

Fresh kefir grains were washed three times using deionized water and then placed in a glass jar covered with a breathable cloth, so that excess carbon dioxide (CO_2_) produced during fermentation escapes. They were grown in milk, without agitation, under aerobic conditions at room temperature, as higher temperatures lead to a decrease in kefiran production [20]. In this study, three different types of milk, buffalo, cow, and goat milk, were used. The medium was changed daily. These steps were repeated for a few days until the grains had fully acclimated to their new environment, which was when they were able to thicken the milk in 24 h. This can take anywhere from 7–10 days to ensure the grain viability. The kefir grains were separated from the fermented milk (kefir) using a plastic sieve and washed with water.

To obtain the total EPS content [19], 10 g of activated kefir grains were cultured at 20–25 °C in three different types of animal milk (cow (A), buffalo (B), and goat (C)), in a previously sterilized glass container (in autoclave at 134 °C and p = 2.1 bar) covered with a breathable cloth. After 24 h, the grains were separated by filtration.

### 2.3. pH and Growth of Kefir Grains

The milk pH was measured for all the samples after 24 h of fermentation with a previously calibrated pH-meter electrode (Hanna Instruments, Padua, Italy). The grains were weighed with an analytical balance after centrifugation at 3000 rpm (t = 10 min, T = 4 °C) to remove milk residue.

The kefir grain growth was defined as the increase of the kefir grain wet mass after 24 h compared with that at the start of the fermentation, divided by the kefir grain wet mass at the start of the fermentation, and expressed as percentage (%, *w*/*w*):Grain growth % = (w_f_ − w_0_)/w_0_ ∗ 100
where w_f_ is the weight of kefir grains at the start of the fermentation and w_0_ is the weight of kefir grains after 24 h of fermentation.

### 2.4. Bacterial and Yeast Counts

Microbiological analysis to study the effect of the fermentation on the cell counts of the LAB and yeast within the kefir grains was carried out in duplicate according to a known protocol [21].

The kefir grains (1 g) were homogenized in sterile saline water (NaCl 1:10,000 *w*/*v*) and the solutions were centrifuged (10,000 rpm 5 min, 20 °C) to precipitate the cells. Serial diluted solutions were plated on sterile bromocresol purple (BCP) agar (Condalab, Madrid, Spain) at 37 °C for three days and on sterile potato dextrose agar (Condalab, Madrid, Spain) at 24 °C for five days to evaluate total bacterial and yeast counts, respectively. The results were expressed as CFU per mL of sample ± standard deviation.

### 2.5. Extraction and Isolation of Kefiran

Kefiran from A, B and C grains was extracted using three different methods:
(I)*Cold method*: Grains were placed in a flask containing distilled water (*w*/*v* = 1/10) and left under magnetic stirring at room temperature for 1 h.(II)*Hot method*: Grains were placed in a flask containing water (*w*/*v* = 1/10) and left under magnetic stirring in a heating bath at 90 °C for 1 h.(III)*Mild heat + Ultrasonic method*: kefir grains were immersed in a beaker containing mildly heated water at 65 °C (*w*/*v* = 1/10) and then sonicated for 10 min using a frequency of 24 kHz and power of 100 W (Ultrasonic Processor Hielscher—Model UP400S, Teltow, Brandburg, Germany). In the three methods, treatment temperatures should not exceed 90 °C as the polymer structure may denature at 100 °C, as reported by Pop et al. [22].

Following the initial treatment, samples from all three methods (I, II and III) were processed individually, as follows: First, the sample was centrifuged (Model 2-16 KL, Sigma Laborzentrifugen GmbH, Germany) at 10,000 rpm for 15 min. The supernatant was added to warmed ethanol at −20 °C at a ratio of 1:1 (1:1, *v*/*v*) and stored at −20 °C overnight, to precipitate exopolysaccharides in the mixture. The solid was separated by centrifugation (10,000 rpm, t = 20 min, T = 4 °C), re-dissolved in distilled water, and re-centrifuged under the previous conditions. This treatment was carried out three times to obtain a white sediment, which was freeze dried (Telstar Lyoquest, Terrassa, Barcelona, Spain) [23,24].

### 2.6. Exopolysaccharide Characterization

#### 2.6.1. Determination of Protein Content Using a Bradford Assay

The Bradford test was the simplest and fastest spectrophotometric method for the quantification of total protein. Briefly, 10 μL of the polymer solution (10 mg mL^−1^) was added to 100 μL of Coomassie Brilliant blue reagent after previous dilution with water (1:5). After an incubation period of 5 min at room temperature, the resultant blue colour was measured at 595 nm (Ultrospec 2100 Pro, Amersham Biosciences/GE Healthcare, Marseille, France). The protein concentration of the sample was determined by constructing a calibration curve of the standard, bovine serum albumin (BSA), in the range of 100–25 μg mL^−1^.

#### 2.6.2. Determination of Total Sugar Content Using a Phenol-Sulphuric Acid Assay

The total sugar content was estimated using the phenol-sulfuric acid colorimetric method [25] with glucose as the standard in the range of 0.1–0.005 mg mL^−1^ (y = 13.346x + 0.0068, R^2^ = 0.9983). Briefly, the polymer solution (10 µL, 1 mg mL^−1^) was added to a phenol solution (1 mL, 6% *w*/*v*) and sulfuric acid (5 mL) for 5 min at room temperature, followed by heating the solution in a boiling water bath for 15 min under magnetic stirring. The absorbance was measured at 490 nm with a UV-Vis spectrophotometer against a blank containing 10 µL of water, 1 mL of phenol solution and 5 mL of sulfuric acid. The analyses were performed in triplicate.

#### 2.6.3. Differential Scanning Calorimetry (DSC)

The thermal properties of the samples (3–10 mg) were ascertained with a DSC SETARAM 131 instrument. Analyses were performed from −20 °C to 500 °C at a temperature scan rate of 20 °C min^−1^ under nitrogen flow [26].

#### 2.6.4. Morphological Characterization (SEM)

Morphological features of all the EPSs were characterized using a scanning electron microscope (SEM) (Field Emission SEM FEI Quanta 200, Thermo Fisher Scientific, Hillsboro, OR, USA) at 15 KV. All the samples were coated with a 5-nm layer of carbon, using a QUORUM Q150T-ES Carbon Coater (Darmstadt, Germany). Morphological images were acquired from the scattered electron signal (SE signal) and the crystal characteristics were observed with the electron technique (BSE signal).

#### 2.6.5. Infrared Characterization (FT-IR)

All extracts obtained were characterized, based on their fingerprint, by FT-IR spectroscopy using a Bruker ALPHA FT-IR spectrometer (Billerica, MA, USA) equipped with an A241/D reflection module. The pulverized samples were prepared by mixing them with KBr and compressing the resulting powder with a press at a force of 6 tons [27]. The spectra were recorded in the wavelength range of 4000–400 cm^−1^.

### 2.7. Statistical Analysis

The experimental data obtained for pH values, grain growth, LAB and yeast counts and recovery of kefiran from the three methods, were expressed as means of three replicates ± standard deviation. The statistical analyses were performed using GraphPad Prism 8.0.2 software (GraphPad Software, Inc., San Diego, CA 92121 USA), and evaluated by two-way ANOVA followed by a Sidak test to make multiple comparisons for pH and grain growth, and by the Tukey method to evaluate the significance of different recovery amounts of kefirans and microbial counts. Significance was established at *p* values < 0.05 (*), *p* < 0.01 (**), *p* < 0.001 (***), and *p* < 0.0001 (****).

## 3. Results

### 3.1. pH and Growth of Kefir Grains

At time zero, before inoculation with starter cultures, the pH values were slightly different depending on the origin of the milk, i.e., the pH values of buffalo, cow, and goat milk were close to neutral (6.60 ± 0.2, 6.70 ± 0.1, and 6.89 ± 0.1, respectively). After 24 h of fermentation, the pH values of buffalo, cow, and goat milk had decreased significantly (3.5 ± 0.1, 3.4 ± 0.2, and 3.6 ± 0.1, respectively) from the initial value due to the fermentation of lactose by the LAB producing lactic acid (Figure 1) [28,29].

The LAB and yeast from the Kefir grains produced an exopolysaccharide matrix called kefiran [5] resulting in an increase in the final weight of the grains, which confirmed the amplified activity of the bacteria. The data obtained (Figure 2), showed that the highest increase in biomass was for grains grown in cow milk, whose weight doubled after 24 h (grain growth % = 100), followed by the grains in buffalo milk and then goat milk, whose weight after fermentation was 33.5 ± 0.1 g (grain growth % = 67.5) and 30.7 ± 0.2 g (grain growth % = 53.5), respectively. However, grain growth was significant in all three types of milk (**** *p* < 0.001).

### 3.2. LAB and Yeast Content

The microbial composition of kefir grains after 24 h of fermentation highlighted different LAB and yeast populations depending on the type of milk used (Figure 3). Cow milk kefir grains showed the highest concentration of LAB (9.03 ± 0.27 × 10^8^ CFU mL^−^^1^) compared with those of buffalo and goat milk (5.13 ± 0.35 × 10^8^ and 4.53 ± 0.25 × 10^8^ CFU mL^−1^, respectively). No significant difference in the growth of the LAB population between buffalo and goat milk grains was found. The yeast population of goat milk grains (6.76 ± 0.25 × 10^8^ CFU mL^−1^) was significantly higher than those of cow and buffalo milk grains (2.93 ± 0.75 × 10^8^ and 2.93 ± 0.75 × 10^8^ CFU mL^−1^, respectively).

### 3.3. Extraction Yields

The extraction yields of kefiran from kefir grains processed using the three methods I, II, II are shown in Figure 4. The lowest yield was observed for the cold treated samples, ranging from 0.15 ± 0.02% for goat milk to 0.48 ± 0.05 for buffalo milk, while the high temperatures led to significantly higher yields (2.80 ± 0.01–3.04 ± 0.01%). The combined method resulted in recovery of kefiran in yields comparable to those obtained with the hot method (* *p* < 0.05 for buffalo kefiran, ** *p* < 0.01 for goat kefiran), except for the cow milk sample from which the highest amount of exopolysaccharide was extracted (4.79 ± 0.01%, **** *p* < 0.0001), compared with all other samples and methods.

### 3.4. Determination of Protein and Total Sugar Content

The protein content of the obtained polymer was different depending on the extraction method (Figure 5A). The cold method led to the highest protein content for all milk samples (1.9 ± 0.1%, 1.2 ± 0.1%, 1.4 ± 0.2%, from buffalo, cow, and goat milks, respectively), except for kefiran from buffalo milk extracted with hot water, which was recovered in a similar percentage (1.4 ± 0.1%). In all other samples, methods II and III led to an exopolysaccharide in which proteins were absent, with the only exception being where kefiran was extracted from goat milk grains, and then the protein content was 0.28 ± 0.03%. Total sugar concentration was found to be high in all kefirans (Figure 5B). The results showed that the percentage of sugar in the kefiran extracted with cold water (I) ranged from 97.2 ± 0.4 for the buffalo milk sample to 98.2 ± 0.6 for the cow polymer. Exopolysaccharides from methods II and III contained the highest percentage of sugars (99.1 ± 0.9–99.8 ± 0.2), except for the kefiran extracted with method III from goat milk, whose percentage of sugars was the lowest, compared to the other samples (92.0 ± 1.2).

### 3.5. DSC

Differential scanning calorimetry was employed to evaluate the thermal properties and transitions of all the kefirans [30]. The thermal transitions of cow, buffalo, and goat kefirans, are reported in Table 1 and the corresponding DSC thermograms are shown in Figure 6. The calorimetric curves of all kefiran samples showed two main peak transitions, both endothermic peaks, one in the 91.7–162.2 °C range and one between 263.6 °C and 353.8 °C. In particular, the first peak could be associated to the melting point of the polymer due to water bound to the OH functional groups of the structure [31], while the second one could be caused by the degradation of kefiran. Such a wide range of melting and degradation temperatures for kefiran samples is related to the type of milk and to the extraction method used. Cow kefiran obtained using method I exhibited an inflection point at 95.7 °C, while the ones extracted with methods II and III showed sharp and broad endothermic peaks above 100 °C (109.1 and 124.8 °C, respectively). Furthermore, the highest melting enthalpy was associated with the AII sample (195.8 J/g). The second peak transition was sharper than the first and varied by only 20 °C depending on the extraction method, from 263.6 °C for AI to 284.2 °C for AIII. The AI enthalpy of degradation was the lowest one (40.0 J/g), while the AIII degradation enthalpy was the highest (168.4 J/g). The two transitions of BI were not represented by sharp peaks, and they were aligned to 124.6 and 294.5 °C. The BII and BIII samples highlighted the highest T_m_ (162.2 °C) and T_d_ (323.7 °C), respectively. The calorimetric curve of goat kefiran (CI) extracted using the method I was different from that of cow and buffalo kefiran extracted using the same method (AI and BI, respectively). Two inflection points were identified, corresponding to T_m_ and T_d_ at 91.7 °C and 353.8 °C, respectively. The calorimetric curves of goat kefiran extracted using methods I and II were similar in the regions of the transition peaks related to T_m_ (142.6 °C for CII and 132.2 °C for CIII) and T_d_ ranged from 277.2 °C to 283.3 °C.

### 3.6. SEM

Scanning Electron Microscopy is useful for studying the surface topography of materials [32]. Each sample was analysed at three magnifications, equal to 100X, 1000X and 10,000X, with a scale of 1.0, 100.0 and 10.0 µm, respectively. All the kefiran images are presented in Figure 7. The morphological analysis of cow kefirans (Figure 7A) exhibited a filamentous and vesicular structure; in addition, the surface of AI showed rounded microstructural vesicles. The difference between the surfaces of the kefiran from the three different extraction methods were due to the size of the vesicles, which increased from 6–9 µm in AI, to 20–40 µm in AII, up to 18–85 µm in AIII. Buffalo kefiran (Figure 7B) extracted in hot water (BII) showed a compact surface with a filamentous and vesicular structure. The size of vesicles ranged from 20 to 40 µm. Kefiran obtained using the ultrasonic method (BIII) showed a filamentous structure with some macro-vesicles (18–85 µm). All samples B had a filamentous structure with vesicles. They differed in the size of the vesicles, which increased from about 20–40 µm in sample BI to 18–85 µm in sample BIII. The morphological analysis of buffalo kefiran (Figure 7B) extracted with cold water (I) exhibited a compact granular structure with fragments immersed in a matrix probably of polymeric nature. Its surface was irregular and had protuberances, and no vesicles were present. The morphological characteristics of kefirans extracted with hot water (II) and mild heat-ultrasound (III) were very similar showing a lamellar structure (sheeting structure) with the presence of macro-vesicles; both surfaces differed only in the size of the vesicles that were of the order of 80–160 µm for BII and 22–48 µm for BIII. Figure 7C showed the surface morphology of goat kefirans. The ones extracted with cold water (CI) showed a sheeting structure (leafy), with irregular fragments. The surface was not smooth but rather was rippled and had a hint of porosity. In fact, a few pores with dimensions between 1–3 µm were detected. Kefiran obtained from hot water (CII) had a compact lamellar structure with some macro-vesicles (50–300 µm). It also had a nano-porosity with the diameter of pores 200–500 µm. Sample CIII had a compact structure, that was intermediate between filamentous and laminar ones. Unlike sample CII, its surface did not show any porosity but only vesicles (5–35 µm).

### 3.7. FT-IR

Infrared spectroscopy of all samples was carried out to further characterize the kefirans and to identify the fundamental groups of the polymer. The IR spectra obtained from all the polymers showed no significant differences in their qualitative profiles as they exhibited the characteristic bands of the kefiran [33] (Figure 8). The stretching of the O-H groups in the constituent sugar residues formed an intense peak around 3450–3291 cm^−^^1^. Two weak bands at 2923–2914 cm^−1^ and 2853–2814 cm^−1^ were associated with the vibrations of C-H in sugar rings. The peak around 1656–1634 cm^−1^ was formed from the bending vibration of water molecules trapped in the polysaccharide matrix. In the 1200–1000 cm^−1^ range, the stretching of C-O-C and C-O bonds in the ring formed two strong bands at 1076 and 1025 cm^−1^, which confirmed the structure of the polysaccharide [34]. In the so-called “anomeric region” ranging from 950 to 780 cm^−1^, characteristic weak absorptions at 950–892 cm^−^^1^ and 870–830 cm^−1^ were present indicating α- and β- configuration, respectively, of glycosidic bonds between monomeric units [31,35]. The spectra of the cow, buffalo, and goat kefirans extracted using method I showed deficiencies relative to the fingerprint region (1200–750 cm^−1^). On the other hand, the same spectra showed the presence of a significant band at 1745 cm^−1^ whose intensity was more pronounced in both the samples B and C, suggesting that glucuronic acid or diacyl ester was present [32].

## 4. Discussion

Kefiran was extracted using two traditional methods involving water as the extraction solvent at different temperatures (25 °C and 90 °C, method I and II, respectively) and one advanced methodology performed with ultrasound and a mild heat temperature (65 °C) [19]. The three methodologies were tested on three types of animal milk (cow milk indicated by A, buffalo milk indicated by B, goat milk indicated by C) to assess the influence of the nature of the dairy matrix on the characteristics of the biopolymer of interest. The results showed that the kefiran yield varied depending on the type of milk and on the extraction method. The lowest percentages (less than 0.5%) were recorded for all polymers extracted at room temperature: goat kefiran was recovered with 0.15 ± 0.02% yield, while buffalo kefiran was three times more (0.48 ± 0.05%). The use of high extraction temperatures improved the yield in all three milks, as higher temperatures facilitate the penetration of water into the grains, increasing the solubilisation of the exopolysaccharide and the extraction yield [36]. On the other hand, high temperatures could cause thermal degradation of the biopolymer leading to changes in its actual structure. For this reason, it was decided to combine mild temperatures with the use of ultrasound to reduce the extraction times. Vilkhu et al. [37] found that sonication at 68 °C for 32 min increased the yield by 1.5 times compared to that of heat treatment alone (100 °C for 120 min) [37]. Nevertheless, it can be observed from the results that, despite the use of lower temperatures (65 °C), accompanied by a considerable reduction in time (10 min), the kefiran yields were similar to those obtained with the hot method (* *p* < 0.05 and ** *p* < 0.01) that was statistically confirmed to be the most valid (**** *p* < 0.0001), with the exception of the cow kefiran recovery, which was found to be the highest among all milks and methods used [38]. Extraction yields of cow kefiran as influenced by different extraction methods were reported by Hasheminya et al. [19], highlighting lower kefiran recovery (1.77 ± 0.19) by using the hot method and comparable yield under ultrasound extraction (4.32 ± 0.04) [19]. The pH values in all three milks were similar between 3.4 to 3.6, indicating the high production of lactic acid by the LAB during 24 h of fermentation. These results agree with those of Zajšek et al. [6] highlighting that the growth curve of symbiotic microorganisms in Kefir grains reached a stationary phase between 30 and 40 h of fermentation after which they started to decrease (death phase). In turn, these microorganisms may produce metabolic products that estimate each other’s growth causing an increase in biomass. Along with the growth of microorganism and biomass, there was a gradual decrease in pH values to about 3.5, due to the activity of the bacteria that convert glucose, resulting from the hydrolysis of galactose. In the tested milks, the minimum pH threshold was already reached after 24 h of fermentation. In particular, the lowest pH value related to cow milk (3.4 ± 0.2), indicated the high production of lactic acid by the LAB during 24 h of fermentation (Figure 1). The highest LAB population in kefir grains from cow milk, rather than from buffalo and goat (Figure 3), would explain the greater biomass production (Figure 2) during fermentation as an increase in grain weight as a consequence of the bacterial activity. The growth of the grains also agreed with the extraction yields in the three types of milk, as EPS are biological polymers secreted by the LAB, mainly *Lactobacillus kefiranofaciens*, to create a physical barrier on the cell surface protecting from stress conditions induced by temperature, pH, antibiotics, host immune defence [39,40]. The growth of grains in cow milk was to twice the weight of the starting grains, explaining the greater recovery of kefiran. In addition, the increase was higher than that obtained by Zajšek et al. [6] from the initial value of 42 g/L to the final value of 56.5 g/L.

The presence of low amounts of protein in the composition of kefiran samples was investigated; the protein is considered as an impurity of the exopolysaccharide. It was found that the percentage of protein in the kefiran samples was either 0 %, or reached a maximum of 1.2–1.4% (AI, BI, CI and AII), i.e., using low extraction temperatures, the content of contaminating proteins was higher than that obtained with the two methods employing temperatures above 60 °C. Similar studies have reported that the protein content of the kefiran extracted using high temperature was around 0.01% [11,41,42]. The low extraction temperature used in method I inhibited the denaturation of proteins, thereby favouring the retention of the proteins in the structure, which otherwise would be denatured by using methods II and III. The lower protein content of the heat-treated samples compared to the cold method can precisely be attributed to the thermal denaturation of proteins [41]. The only exception was the protein content of kefiran from buffalo milk, which was high despite the use of high extraction temperatures. This discrepant result was probably due to the higher concentration of fat in buffalo milk compared to cow and goat milk, which, during extraction, could be distributed in the matrix in such a way as to inhibit the denaturation and subsequent separation of the proteins. The total sugar content extracted using the three different methods, even for the ultrasound treatment, did not exhibit significant differences, in accordance with the results obtained by Tang et al. [43]. It was not the treatment with ultrasound, but the effect of high temperatures that determined a greater dissolution and diffusivity of polysaccharides in water. Nevertheless, the temperature should not be too high because it could lead to structural degradation of the polymers [44]. As the applicability of kefirans in the pharmaceutical and food fields is largely dependent on their physical behaviour, the extracted polymers were characterized by SEM, DSC and FT-IR to identify any quali-quantitative differences in the bands characterizing the polymer as a function of the extraction methods [31]. The SEM images of all the samples highlighted that the different milk origins and extraction methods most likely induced changes on the surface morphology, which was clearly dissimilar for all the kefirans [45,46]. However, the obtained SEM images showed that all kefirans were highly porous materials, which are structures having high water holding capacity, so they could find application in the food industry. Additionally, SEM images indicated that the morphological features of the BI surface were comparable to that of exopolysaccharide from human breast milk produced by *Lactobacillus plantarum* HM47, although it had been grown in a different environment condition [47], but none of the kefiran samples were comparable to the cow EPS produced by *Streptococcus thermophilus* that has irregular lumps with a coarse surface [48]. The morphology of AII was very similar to the porous one of an EPS isolated using the hot water extraction from Turkey kefir grains cultured in ultrahigh temperature (UHT) treated skimmed milk [17]. Thermal analyses of all the kefiran samples were carried out to evaluate their transition temperatures, such as melting (T_m_) and degradation (T_d_) temperatures, and the related enthalpies (ΔH_m_ and ΔH_d_). The results highlighted that the thermal features varied in all samples depending on both milk types and the extraction methods (Figure 6 and Table 1). The melting temperature was determined by the loss of bound water and depended on the method of extraction and on the type of milk. It was lower in all the kefirans isolated using method I, except for BI, which had a melting temperature of 125 °C, probably due to a high percentage of fats that could inhibit the loss of water during thermal analysis. Except for AI and CI, all other samples exhibited a higher T_m_ than that of the previously studied kefirans, which turned out to be below 100 °C. Earlier studies demonstrated that the EPS isolated from *L. plantarum* KF5 exhibited a melting point at 88.35 °C [31]. The thermal degradation of polysaccharide probably occurred through a process involving desorption of physically absorbed water, removal of structural water, depolymerization and degradation of monosaccharide leading to production of CO, CO_2_, and H_2_O, and, finally, formation of polynuclear aromatic structures [49]. All isolated kefirans showed a thermal stability higher than 260 °C with a T_d_ between 264 and 295 °C, depending on the extraction method and the type of milk, according to data reported in the literature. For example, the thermal stability (T_d_) of EPS produced by *Lactobacillus plantarum* HM47 isolated from human breast milk was found to be up to 273.6 °C [33,47]. Among all the kefiran samples, the CI had a very high thermal stability, above 350 °C. The thermal stability of the EPSs makes them promising additives for food industries, such as gelling and thickening agents. The IR spectra obtained for kefiran samples from each type of milk, regardless of the method used, showed no significant differences in the qualitative profile of the bands characterizing the polymer structure, except for their intensities. Even the treatment with ultrasound did not lead to changes in the functional groups of kefiran under study, confirming the typical structure of a polysaccharide [43].

## 5. Conclusions

The results showed that the simultaneous use of ultrasound and mildly heated water increased the extraction yield of cow kefiran, while the recovery of buffalo and goat kefirans were unaltered compared to the hot water treatment when applied alone, confirming the hypothesis that the yield of kefiran extraction depends on the type of milk. Unlike method I, methods II and III did not result in significant changes in total sugar and protein contents. Possible differences regarding the studied physicochemical characteristics were also evaluated. The analysis using infrared spectroscopy highlighted that the three different methods did not cause significant changes in the primary chemical structure of polysaccharides, which showed the same qualitative profile of the bands characterizing the functional groups of the structure. Unlike the spectroscopic characteristics, the thermal and morphological features of the samples showed differences according to both the type of the milk and the extraction method. However, to fully understand the effect of these morphological and thermal differences on the practical applications of kefiran in the fields of food and pharmacology, biochemical investigations will be conducted.

## Figures and Tables

**Figure 1 foods-11-01098-f001:**
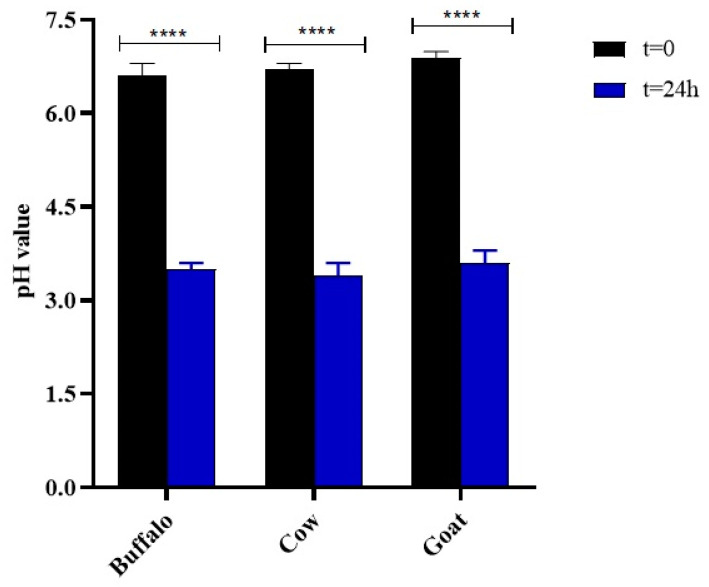
pH values of the samples at 0 h and 24 h of fermentation and their relative significance. Histograms and error bars represent the means of three independent experiments and standard deviations, respectively. Asterisks on bars indicated that the mean values were statistically different from time zero (**** *p* < 0.0001).

**Figure 2 foods-11-01098-f002:**
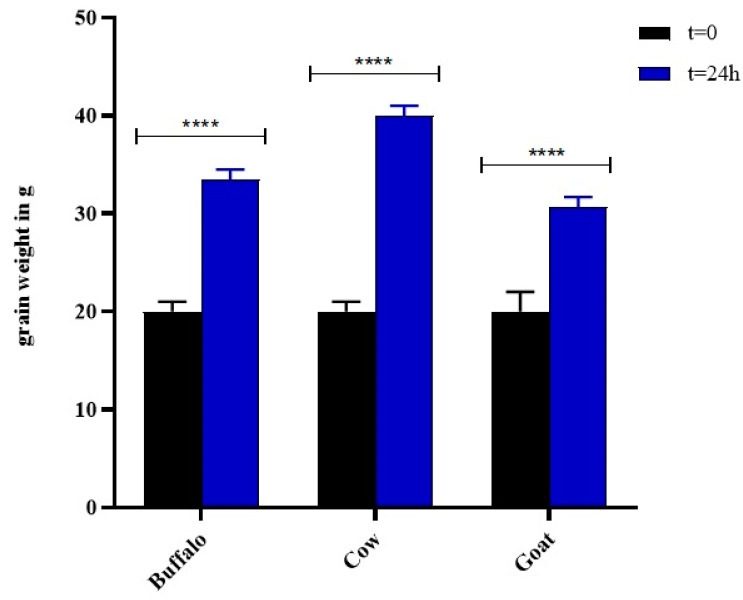
Mean values of grain weight ± standard deviation at 0 h and 24 h of fermentation time. Histograms and error bars represent the mean of independent experiments performed in triplicate and standard deviation, respectively. Asterisks on bars indicated that mean values were statistically different from time zero (**** *p* < 0.0001).

**Figure 3 foods-11-01098-f003:**
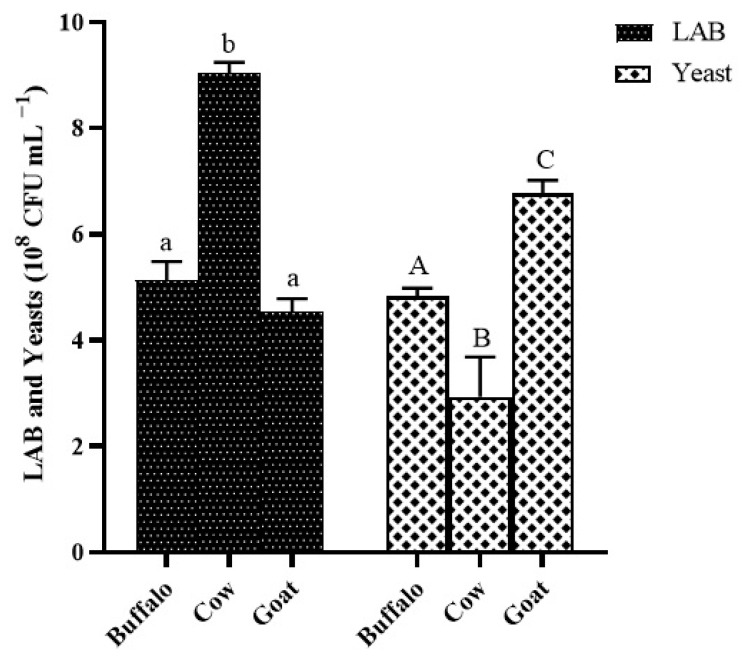
Cell counts of lactic acid bacteria and yeast populations in kefir grains from fermented buffalo, cow, and goat milk after 24 h. Histograms represent the averages of three independent experiments (each done in triplicate) and error bars indicate the standard deviation. Letters on histograms denote statistical analysis. Values with the same letter are not significantly different, values with different letters are significantly different (**LAB**: buffalo vs. cow, cow vs. goat; **Yeast**: buffalo vs. cow, buffalo vs. goat, cow vs. goat).

**Figure 4 foods-11-01098-f004:**
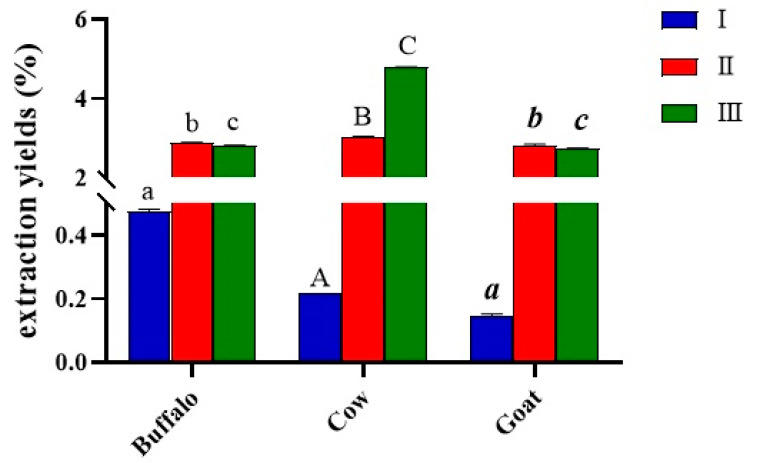
Extraction yields of kefiran, expressed as %, as influenced by the different extraction methods of using cold water (I), hot water (II), and a hybrid mild heat-ultrasound (III) method. The letters on the histograms indicate the statistical analysis. Values with the same letter are not significantly different; values with different letters are significantly different (**Buffalo**: I vs. II, I vs. III, II vs. III; **Cow**: I vs. II, I vs. III, II vs. III; **Goat**: I vs. II, I vs. III, II vs. III).

**Figure 5 foods-11-01098-f005:**
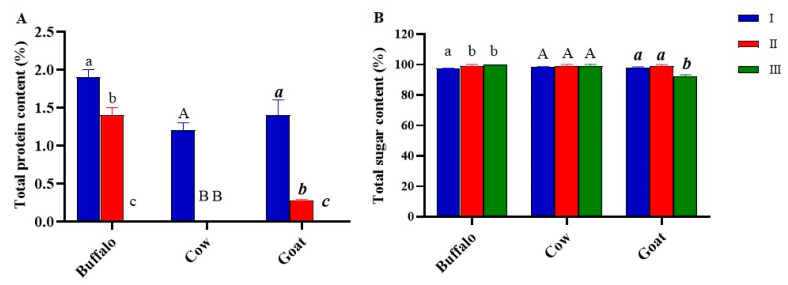
(**A**) Protein content (%) and (**B**) total sugar (%) of kefiran exopolysaccharide as influenced by different extraction methods including cold water (I), hot water (II), and mild heat-ultrasound (III). The letters on the histograms indicate the statistical analysis. Values with the same letter are not significantly different; values with different letters are significantly different. (**A**), **Buffalo**: I vs. II, I vs. III, II vs. III; **Cow**: I vs. II, I vs. III; **Goat**: I vs. II, I vs. III, II vs. III. (**B**), **Buffalo**: I vs. II, I vs. III; **Goat**: I vs. III, II vs. III.

**Figure 6 foods-11-01098-f006:**
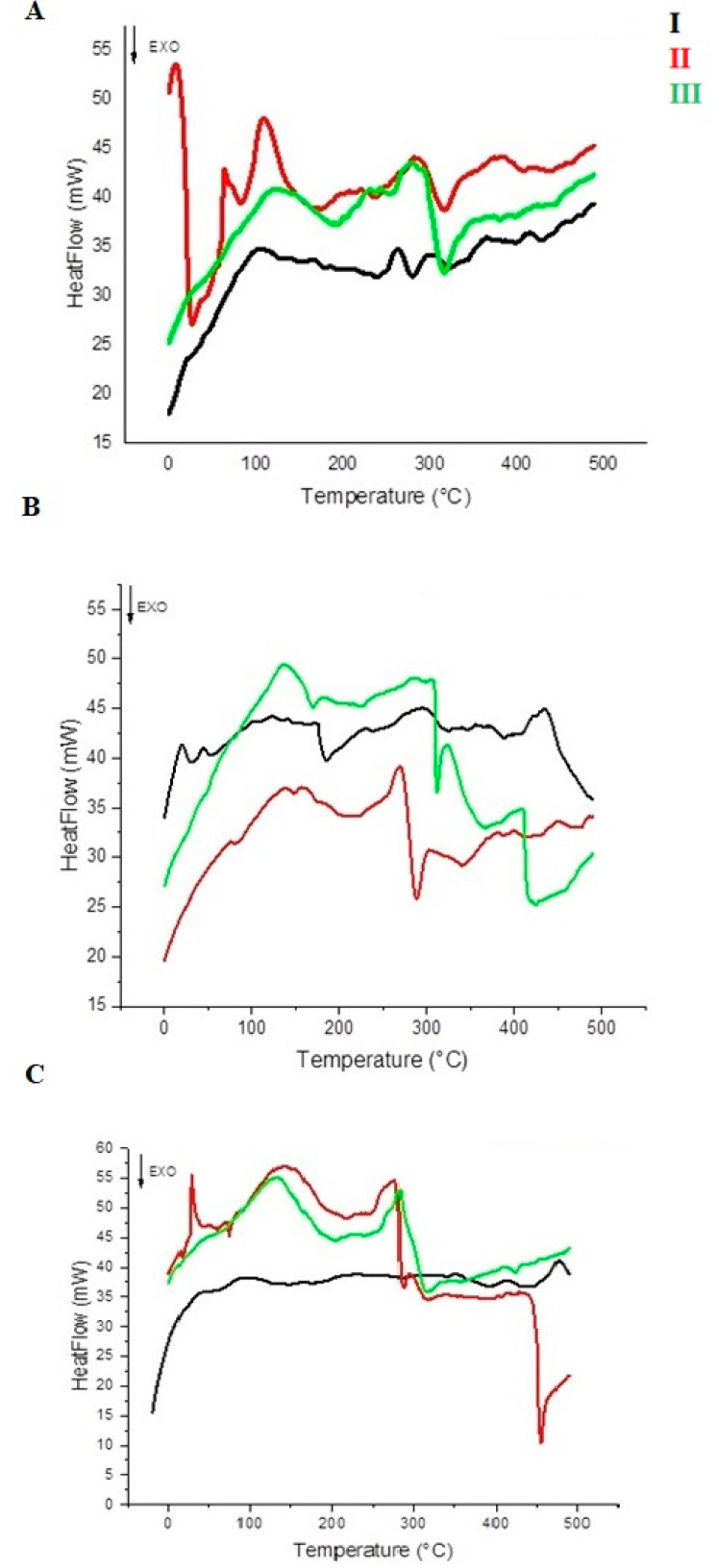
Differential scanning calorimetry of cow (**A**), buffalo (**B**) and goat (**C**) kefirans extracted with cold water (black line), hot water (red line), and mild heat-ultrasound (green line).

**Figure 7 foods-11-01098-f007:**
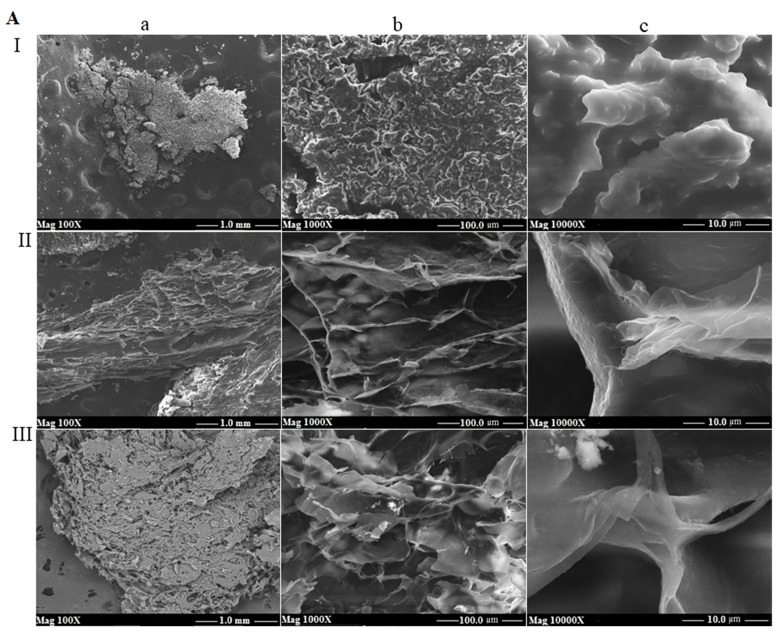
Scanning electron images of cow (**A**), buffalo (**B**), and goat (**C**) kefirans, extracted with cold water (I), hot water (II), and mild heat-ultrasound (III) at three different magnifications and scales: a (100X, 1.0 µm); b (1000X, 100.0 μm); c (10,000X, 10.0 μm).

**Figure 8 foods-11-01098-f008:**
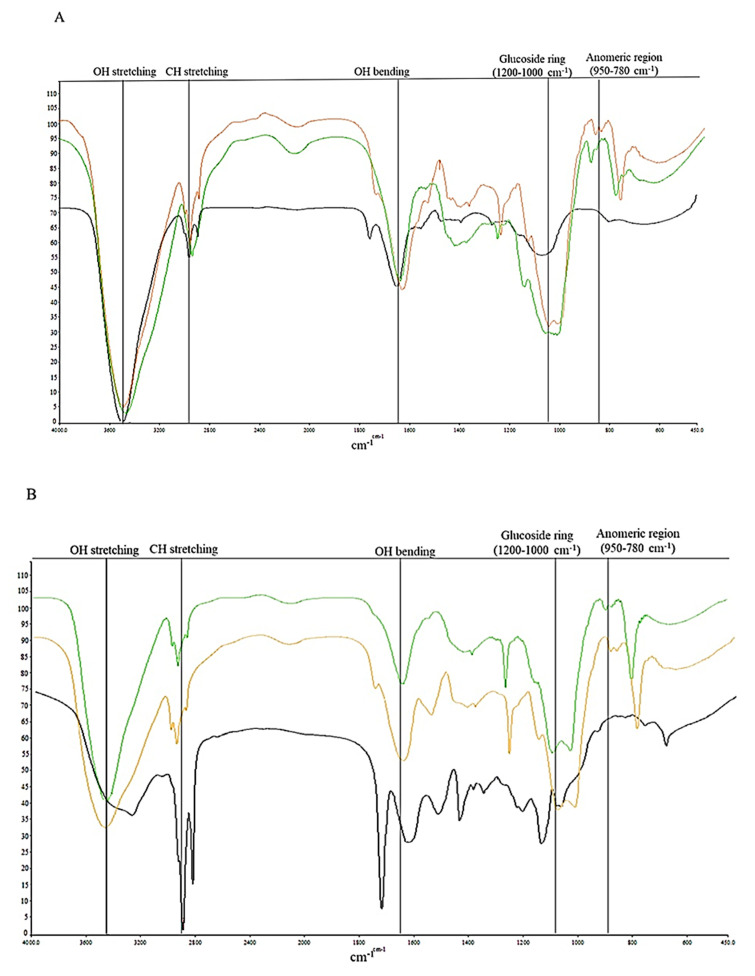
IR spectra of cow (**A**), buffalo (**B**) and goat (**C**) kefirans extracted with cold water (black line), hot water (orange line), and mild heat-ultrasounds (green line) between 4000 and 450 cm ^−1^.

**Table 1 foods-11-01098-t001:** Thermal properties of cow (A), buffalo (B), and goat (C) kefirans extracted with cold water (I), hot water (II), and mild heat-ultrasound (III).

Kefirans	T_m_ (°C)	ΔH_m_ (J/g)	T_d_ (°C)	ΔH_d_ (J/g)
*AI*	95.7	157.6	263.6	40.0
*AII*	109.1	195.8	282.8	135.0
*AIII*	124.8	168	284.2	168.4
*BI*	124.6	45	294.5	35.7
*BII*	162.2	7.3	271.0	109.8
*BIII*	134.8	173.1	323.7	72.4
*CI*	91.7	90.6	353.8	8.4
*CII*	142.6	248.9	277.2	153.8
*CIII*	132.2	323.8	283.3	181.4

T_m_, temperature of melting (°C); ΔH_m_, melting enthalpy (J/g); T_d_, temperature of degradation (°C); ΔH_d_, degradation enthalpy (J/g).

## Data Availability

The data presented in this study are available within the article.

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
