# Peer review of "Influence of Three Extraction Methods on the Physicochemical Properties of Kefirans Isolated from Three Types of Animal Milk"

_foods, 2022, doi:10.3390/foods11081098_

Round 1
Reviewer 1 Report
This manuscript attempted to characterize kefiran extractions from kefir grains carried out with cold water, hot water and hot water-ultrasound. The recovery yield and the physicochemical properties of kefirans were evaluate to establish the influence of both the extraction conditions and the type of milk. The authors provided sight for the kefiran extractions. However, the results do not fully support the conclusion. There are major drawbacks that in my opinion prevent publication in the present form. Discussion of results is confusing, and there are some points that should be added, adjusted and clarified.
Specific comments:
Major comments:
- What is the hypothesis of this study? Is it because the composition of various type of raw milk is different, which will produce different fermentation products? The manuscript should clarify this issue.
- The experimental results show that: the pH of the cow milk fermentation system decreases most significantly in Figure 1; the biomass of cow milk is the highest in Figure 2; and the number of lactic acid bacteria in cow milk is the highest in Figure 3. It is necessary to analyze the relationship between these three results in detail and explore the main influencing factors of the complex fermentation system. Only the total number of colonies is not enough, and the specific microbiota needs to be discussed in detail.
- The results in Fig. 6 and Fig. 8 show that the products under different extraction processes of the same raw milk have significant different profile, the reasons need to be explained.
Minor comments:
- L22: Abbreviations need to be used correctly.
- L35: This reference is not a good example of the manuscript.
- L85: Is the medium sterilized before inoculation?
- L96: “rpm” should be replaced by “g”.
- L101: Please confirm the concentration of sterile saline.
- L114: Cold or Hot?
- L136: Does the lactose content in raw milk influence on the results considered?
- Fig1 is not clear.
Author Response
Reviewer 1
This manuscript attempted to characterize kefiran extractions from kefir grains carried out with cold water, hot water and hot water-ultrasound. The recovery yield and the physicochemical properties of kefirans were evaluate to establish the influence of both the extraction conditions and the type of milk. The authors provided sight for the kefiran extractions. However, the results do not fully support the conclusion. There are major drawbacks that in my opinion prevent publication in the present form. Discussion of results is confusing, and there are some points that should be added, adjusted and clarified.
Response: We thank the reviewer for his/her useful comments to improve our research article.
We revised “Discussion” and “Conclusions” sections as you suggested.
Specific comments:
Major comments:
- What is the hypothesis of this study? Is it because the composition of various type of raw milk is different, which will produce different fermentation products? The manuscript should clarify this issue.
Response: We thank the reviewer for his/her useful question. The hypothesis of this study is that the different composition of various types of raw milk will produce different fermentation products with the same extraction method.
In the “Introduction” section, we have inserted the hypothesis of the study as follows:
“The hypothesis of this study is that the different composition of various type of raw milk could lead to different exopolysaccharide. In order to refute or confirm the hypothesis, three types of animal milk (cow, buffalo and goat) were used for the growing of kefir grains from which to extract microbial exopolysaccharides by three methods.”
- The experimental results show that: the pH of the cow milk fermentation system decreases most significantly in Figure 1; the biomass of cow milk is the highest in Figure 2; and the number of lactic acid bacteria in cow milk is the highest in Figure 3. It is necessary to analyze the relationship between these three results in detail and explore the main influencing factors of the complex fermentation system. Only the total number of colonies is not enough, and the specific microbiota needs to be discussed in detail.
Response: We thank the reviewer for his/her suggestion. We have discussed the relationship between the pH values, LAB population counts and grain growth in the “Discussion” section as follows:
“The pH values in all the three milks were similar ranging between 3.4 to 3.6 indicating the high production of lactic acid by the LAB during 24 h of fermentation. These results agree with the ones from the study of Zajšek et al. [6] highlighting that the growth curve of symbiotic microorganisms in Kefir grains reached a stationary phase between 30 and 40 h of fermentation after which they started to decrease (death phase). In turn, these microorganisms may produce metabolic products that stimulate each other’s growth causing an increase in biomass. Along with the growth of microorganism and biomass, there was a gradual decrease in pH values to about 3.5, due to the activity of the bacteria that convert glucose, resulting from the hydrolysis of galactose, into lactic acid. In the tested milks, the minimum pH threshold was already reached after 24 h of fermentation. The lowest pH value related to cow milk (3.4 ± 0.2), indicated the high production of lactic acid by the LAB during 24 h of fermentation (Fig. 1). The highest LAB population in kefir grains from cow milk, rather than from buffalo and goat (Fig. 3), would explain the greater biomass production (Fig. 2) during fermentation since grains weight increment is a consequence of the bacterial activity.”
We are aware that the specific microbiota is an important factor in the development of fermentations that we will consider and study in a separate future manuscript, but, at this stage, the focus of our study was the influence of extraction methods on the characteristics of kefirans produced in all experiments by the complex of microorganisms coexisting in the same batch of kefir grains. We did not worry about analysing the various populations because it did not represent a variable in the experiments.
- The results in Fig. 6 and Fig. 8 show that the products under different extraction processes of the same raw milk have significant different profile, the reasons need to be explained.
Response: We thank the reviewer for his/her comment.
IR spectra of cow (A), buffalo (B) and goat (C) kefirans reported in Figure 8 extracted with the three different methods under study, showed no significant differences in the qualitative profile of the bands characterizing the polymer structure (stretching of OH groups at 3450-3291 cm-1, CH stretching in sugar rings at 2923-2814 cm−1, stretching of ring bonds at 1200-1000 cm−1 and anomeric region at 950-830 cm-1), except for their intensities.
Differential scanning calorimetry thermograms, showed in Figure 6 and discussed in the subsection 3.5. of the “Results” section, highlighted a wide range of melting and degradation temperatures for kefiran samples depending on the type of milk and, within the same one, on the extraction method used.
In the “Discussion” section, we inserted: “The results highlighted that thermal features varied in all samples depending on both milk types and extraction methods (Fig.6 and Table 1).”
Minor comments:
- L22: Abbreviations need to be used correctly.
Response: We thank the reviewer for his/her comment. Abbreviations are correct and listed at the end of the manuscript according to the indications provided by MDPI.
- L35: This reference is not a good example of the manuscript.
Response: We thank the reviewer for his/her comment. In the introduction we have included a brief mention of the importance of the health properties of fermented beverages on the human immune system and consequently of the high consumption of kombucha and kefir during the COVID-19 pandemic. We thought it was important for us to include our publication on the topic.
- L85: Is the medium sterilized before inoculation?
Response: We thank the reviewer for his/her question.
Commercial cow and goat milks were ultra-high temperature-treated milks and for this reason they were not sterilized whereas fresh buffalo milk was used after pasteurization at 90 °C for 15 minutes. In the subsection 2.1. of “Material and methods” section we have inserted complete informations on the used milks as follows:
UHT cow milk, UHT goat milk and fresh buffalo milk and fresh buffalo milk that has been pasteurized at 90 °C for 15 minutes before use.
- L96: “rpm” should be replaced by “g”.
Response: We thank the reviewer for his/her comment.
We have corrected all the entries related to the centrifuge correctly replacing rpm instead of g inserted by mistake.
- L101: Please confirm the concentration of sterile saline.
Response: We confirm
- L114: Cold or Hot?
Response: We thank the reviewer for his/her question.
We have made the correction by replacing “cold” with “mild heat” because 65°C is mild temperature not a high temperature
- L136: Does the lactose content in raw milk influence on the results considered?
Response: We thank the reviewer for his/her question.
The lactose content in raw milk doesn’t influence our results because lactose content does not vary significantly among the three types of milk (5.0 g/100g for cow milk, 5.1 g/100 g for buffalo milk and 4.7 g/100 g for goat milk). In general, it must be considered that symbiotic microorganisms present in kefir grains are homofermentative and heterofermentative lactic acid bacteria, lactose-assimilating and non-lactose-assimilating yeasts. During fermentation homofermentative LABs transform lactose into lactic acid and heterofermentative LABs produce formic, lactic and acetic acids, ethanol and carbon dioxide. Excess lactic acid was consumed, along with galactose, by non-lactose-assimilating yeasts as a carbon source and energy source, but some yeasts are proteolytic and lipolytic and they may obtain cell growth components by this metabolism.
In any way we also studied the fermentation of vegetable milks, which don’t contain lactose but are added with cane sugar (3.0-4.1 g/100g of cane sugar) and we observed that when they were fermented with the same milk kefir grains, they led to even better yields of kefiran. These results suggested that the presence of simple carbohydrates, not necessarily lactose, is important for the nutrition of symbiotic microorganisms, but does not affect the physicochemical differences of the biopolymers.

Reviewer 2 Report
The paper submitted for publication studies the use of cold water, hot water and ultrasound combined with heat on the extraction of kefirans following fermentation of cow, buffalo and goat milk. In addition to the extraction efficiency, the authors propose an extensive characterization of the kefirans obtained.
Overall I recommend this article for publication with some major revisions:
L114: Change "cold + ultrasonic method" by "Mild heat + ultrasonic method". In general, all over the manuscript, I would recommend saying that 65°C is mild temperature not a high temperature. In food processing it is even considered as low temperature. In addition, it is much milder compare to the 90 °C used in method 2.
Section 2.5: I would add the equation used for calculation of yield and efficiency (grain growth)
L127: I am not sure that because "the Bradford test was the simplest and fastest" method for protein quantification, it is the most appropriate. Why choose this method over Dumas ?
L143: The previous sections (2.5 and 2.6) should also be part of the exopolysaccharide characterization (2.8). Else the authors should distinguish physicochemical characterization and structural characterization.
Section 3.1: I am not sure of what the authors want to highlight with the statistic. Is there a single effect of the milk source? single effect of the time? or interaction between the two parameters? From what I understand, the main effect comes from the time. In that order, the author should either make an average of all three types of milk at t=0, and average at t=24h. Or the authors should put a bar with **** for each category of milk (buffalo, cow and goat).
Similarly for Figure 3, if there is only a simple effect of the time on the grain growth, then it would be more correct to make an average of all types of milk at t=0 and t =24 or make three bars (for each milk type).
Figure 4: Statistics are very confusing. It would be more appropriate in this kind of graphic to put letters attributed to each type of milk instead of bars. The strength of the statistical difference (***, ****) might be mentioned in the text only.
Section 3.3: This section is quite confusing as we are not sure if the authors performed a factorial analysis.
In addition, both table (1) and Figure (4) are not necessary. One of them is enough. I recommend keeping the Figure as it is more visual. However some modifications are required:
Why not using the extraction yields % as in the table?
Statistics represented her suggest a single effect of the extraction method, if this is the case, please adapt the Figure with previous comments (using letters instead of bars, average of type of milk, etc.) If there is a factorial effect then it is important to use letters.
Section 3.4/Figure 5: some important modifications are required
Statistics are missing in the Figure 5 a and b. It is important to check factorial effect.
In addition, even though Method III shows protein value of almost 0%, using an histogram would be more appropriate.
LAstly, if the kefirans extracted from goat with Method 3, only have 92% of polysaccharides and almost no protein, what are the other 7/8%?
L245: delete one "the"
Table 2 should also present statistical analysis for each parameter. The analysis should have been done in triplicates, and therefore value should have standard deviation too.
Figure 8, the scale and legend are barely readable.
Discussion: This section requires important english editing in addition to some modification
L327/328: Favour the use of milk temperature.
L348: The transition between extraction yield and pH value and then grain growth should re-edited. I highly suggest the authors to combine results and discussion, which will make the reading much more enjoyable.
In general I would avoid using the world Hot-treated at the expense of heat-treated
L378, you mention statistics that have not been presented before
Conclusions:
L418: rephrase the first sentence which is 5 lines long!
Author Response
Reviewer 2
The paper submitted for publication studies the use of cold water, hot water and ultrasound combined with heat on the extraction of kefirans following fermentation of cow, buffalo and goat milk. In addition to the extraction efficiency, the authors propose an extensive characterization of the kefirans obtained.
Overall I recommend this article for publication with some major revisions:
- L114: Change "cold + ultrasonic method" by "Mild heat + ultrasonic method". In general, all over the manuscript, I would recommend saying that 65°C is mild temperature not a high temperature. In food processing it is even considered as low temperature. In addition, it is much milder compare to the 90 °C used in method 2.
Response: We thank the reviewer for his/her useful comment to improve our research article. We agree with his/ her comment and we have replaced "cold + ultrasonic method" by "Mild heat + ultrasonic method" in the manuscript.
- Section 2.3: I would add the equation used for calculation of yield and efficiency (grain growth)
Response: We thank the reviewer for his/her suggestion. We have reported in the bar diagram the weight of the grains at time zero before inoculation and after 24 of fermentation. As you suggest we add the equation leaving the diagram unchanged and commenting the grain growth percentage in the manuscript. We added the equation in subsection 2.3 of the “Material and methods” section as follows:
The kefir grain growth was defined as the increase of the kefir grain wet mass after 24 h compared with that at the start of the fermentation, divided by the kefir grain wet mass at the start of the fermentation, and expressed as percentage (%, w/w).
Grain growth %= (wf -w0)/w0 *100
where wf is the weight of kefir grains at the start of the fermentation and w0 is the weight of kefir grains after 24 h of fermentation
- L127: I am not sure that because "the Bradford test was the simplest and fastest" method for protein quantification, it is the most appropriate. Why choose this method over Dumas ?
Response: We thank the reviewer for his/her comment. We used Bradford test to estimate the protein content as suggested by the study of Hasheminya & Dehghannya:
Hasheminya, S.M.; Dehghannya, J. Novel ultrasound-assisted extraction of kefiran biomaterial, a prebiotic exopolysaccharide, and investigation of its physicochemical, antioxidant and antimicrobial properties. Mater. Chem. Phys. 2020, 243, 122645.
- L143: The previous sections (2.5 and 2.6) should also be part of the exopolysaccharide characterization (2.8). Else the authors should distinguish physicochemical characterization and structural characterization.
Response: We thank the reviewer for his/her comment. We decided to include subsections 2.6 and 2.7 as parts of the subsection “Exopolysaccharide characterization” without distinguishing between characterization without distinguishing between physicochemical characterization and structural characterization. The section 2.8. Exopolysaccharide characterization has become 2.6 and the subsections 2.6 and 2.7 has become 2.6.1. and 2.6.2, respectively.
- Section 3.1: I am not sure of what the authors want to highlight with the statistic. Is there a single effect of the milk source? single effect of the time? or interaction between the two parameters? From what I understand, the main effect comes from the time. In that order, the author should either make an average of all three types of milk at t=0, and average at t=24h. Or the authors should put a bar with **** for each category of milk (buffalo, cow and goat).
Response: We thank the reviewer for his/her comment. The main effect came from the time of fermentation for each of the three milks. In fact, the pH values dropped significantly after 24 hours of fermentation in all three types of milk from the values of 6.6-6.9 to 3.4-3.6. The pH values after 24 h of fermentation decreased from the initial values with the same significance (**** p < 0.0001), and for this reason it was incorrectly indicated with just one bar. In Figure 1 we put a bar with **** for each type of milk, as you suggested.
Figure 1. pH values of the samples at 0 h and 24 h of fermentation time. Histograms and error bars represent the mean of independent experiments performed in triplicate and standard deviation, respectively. Asterisks on bars indicated that mean values were statistically different from time zero (**** p < 0.0001).
- Similarly for Figure 3, if there is only a simple effect of the time on the grain growth, then it would be more correct to make an average of all types of milk at t=0 and t =24 or make three bars (for each milk type).
Response: We thank the reviewer for his/her comment. In Figure 2 we put a bar with **** for each type of milk, as you suggested.
Figure 2. Mean values of grain weight ± standard deviation at 0 h and 24 h of fermentation time. Histograms and error bars represent the mean of independent experiments performed in triplicate and standard deviation, respectively. Asterisks on bars indicated that mean values were statistically different from time zero (**** p < 0.0001).
- Figure 4: Statistics are very confusing. It would be more appropriate in this kind of graphic to put letters attributed to each type of milk instead of bars. The strength of the statistical difference (***, ****) might be mentioned in the text only.
Response: We thank the reviewer for his/her comment. We denoted the significance levels in Figure 3 by using the letters as you suggested.
Figure 3. Cell counts of lactic acid bacteria and yeast populations in kefir grains from fermented buffalo, cow and goat milk after 24 h. Histograms represent the averages of three independent experiments (each done in triplicate) and error bars indicate standard deviation. Letters on histograms denote statistical analysis. Values with the same letter are not significantly different, values with different letters are significantly different (LAB: buffalo vs. cow, **** p value < 0.0001, cow vs. goat, **** p value < 0.0001; YEAST: buffalo vs. cow, *** p value < 0.001, buffalo vs. goat, *** p value < 0.001, cow vs. goat, **** p value < 0.0001).
- Section 3.3: This section is quite confusing as we are not sure if the authors performed a factorial analysis.
In addition, both table (1) and Figure (4) are not necessary. One of them is enough. I recommend keeping the Figure as it is more visual. However some modifications are required:
Why not using the extraction yields % as in the table?
Statistics represented her suggest a single effect of the extraction method, if this is the case, please adapt the Figure with previous comments (using letters instead of bars, average of type of milk, etc.) If there is a factorial effect then it is important to use letters.
Response: We thank the reviewer for his/her suggestion. We remove Table1 and kept Figure 4 expressing the results as percentage yield, as you suggested. We denoted the significance levels in Figure 4 by using the letters, considering only the effect of the extraction method on each type of milk.
Figure 4. Extraction yields of kefiran, expressed as %, as influenced by different extraction methods including cold water (I) hot water (II), and mild heat-ultrasound (III). The letters on the histograms indicate the statistical analysis. Values with the same letter are not significantly different, values with different letters are significantly different (Buffalo: I vs. II, **** p < 0.0001, I vs. III, **** p < 0.0001, II vs. III, * p < 0.05; Cow: I vs. II, **** p < 0.0001, I vs. III, **** p < 0.0001, II vs. III, **** p < 0.0001; Goat: I vs. II, **** p < 0.0001, I vs. III, **** p < 0.0001, II vs. III, ** p < 0.01).
- Section 3.4/Figure 5: some important modifications are required
Statistics are missing in the Figure 5 a and b. It is important to check factorial effect.
In addition, even though Method III shows protein value of almost 0%, using an histogram would be more appropriate.
LAstly, if the kefirans extracted from goat with Method 3, only have 92% of polysaccharides and almost no protein, what are the other 7/8%?
Response: We thank the reviewer for his/her suggestion. Figure 5 has been replaced by a bar graph and statistical analysis has been added as you suggested.
Goat kefiran, which is characterized by 92% sugars and no protein, has 7/8% matter that we couldn't identify. We, like you, had the same question and thought that it was probably ascribed to salts or other foreign compounds that were incorporated into the biopolymer matrices during precipitation from ethanol. In fact, the exopolysaccharides were not subjected to dialysis purification, which we are going to adopt in the next studies to compare any differences between the precipitated biopolymer and the dialyzed one.
Figure 5. A) Protein content (%) and B) total sugar (%) of kefiran exopolysaccharide as influenced by different extraction methods including cold water (I) hot water (II), and hybrid hot water-ultrasound (III). The letters on the histograms indicate the statistical analysis. Values with the same letter are not significantly different, values with different letters are significantly different. Fig 5A, Buffalo: I vs. II, **** p < 0.0001, I vs. III, **** p < 0.0001, II vs. III, **** p < 0.0001; Cow: I vs. II, **** p < 0.0001, I vs. III, **** p < 0.0001; Goat: I vs. II, **** p < 0.0001, I vs. III, **** p < 0.0001, II vs. III, ** p < 0.01. Fig. 5B, Buffalo: I vs. II, * p < 0.05, I vs. III, ** p < 0.01; Goat: I vs. III, **** p < 0.0001, II vs. III, **** p < 0.0001.
- L245: delete one "the".
Response: We have done.
- Table 2 should also present statistical analysis for each parameter. The analysis should have been done in triplicates, and therefore value should have standard deviation too.
Response: We thank the reviewer for his/her comment. There are no standard deviations in the Table 2, because the DSC analysis were performed twice and not three times due to kefiran insufficiency, but the results were the same.
- Figure 8, the scale and legend are barely readable.
Response: We thank the reviewer for his/her comment. IR spectra of cow (A), buffalo (B) and goat (C) kefirans have been merged into a single figure and therefore the compacted figure has lost resolution. Following your observation, we split the three figures to improve image quality.
Discussion: This section requires important english editing in addition to some modification
- L327/328: Favour the use of milk temperature.
Response: We thank the reviewer for his/her suggestion. We have specified the temperatures used in the respective methods, as follows: 25 °C and 90 °C, method I and II, respectively).
- L348: The transition between extraction yield and pH value and then grain growth should re-edited. I highly suggest the authors to combine results and discussion, which will make the reading much more enjoyable.
Response: We thank the reviewer for his/her suggestion. At this point it would be troublesome for us to merge results and discussions. We have reformulated the relationship between extraction yield, pH values and grain growth in the “Discussion” section as follows:
“The pH values in all the three milks were similar ranging between 3.4 to 3.6 indicating the high production of lactic acid by the LAB during 24 h of fermentation. These results agree with the ones from the study of Zajšek et al. [6] highlighting that the growth curve of symbiotic microorganisms in Kefir grains reached a stationary phase between 30 and 40 h of fermentation after which they started to decrease (death phase). In turn, these microorganisms may produce metabolic products that stimulate each other’s growth causing an increase in biomass. Along with the growth of microorganism and biomass, there was a gradual decrease in pH values to about 3.5, due to the activity of the bacteria that convert glucose, resulting from the hydrolysis of galactose, into lactic acid. In the tested milks, the minimum pH threshold was already reached after 24 h of fermentation. The lowest pH value related to cow milk (3.4 ± 0.2), indicated the high production of lactic acid by the LAB during 24 h of fermentation (Fig. 1). The highest LAB population in kefir grains from cow milk, rather than from buffalo and goat (Fig. 3), would explain the greater biomass production (Fig. 2) during fermentation since grains weight increment is a consequence of the bacterial activity.”
- In general I would avoid using the world Hot-treated at the expense of heat-treated
Response: We thank the reviewer for his/her comment. We replaced “hot-treated” by “mild-heat”.
- L378, you mention statistics that have not been presented before
Response: We thank the reviewer for his/her comment. Figure 5 has been replaced by a bar graph and statistical analysis has been added as you suggested.
Conclusions:
- L418: rephrase the first sentence which is 5 lines long!
Response: We thank the reviewer for his/her comment. We removed the sentence as suggested by the reviewer 4

Reviewer 3 Report
The authors investigated the effects of using three types of
animal milk (cow, buffalo, and goat) to grow kefir grains from which were used to extract microbial exopolysaccharides. They also characterized the physicochemical properties of the biopolymersin order to establish a preliminary identification, based on their sugar and protein composition, thermal properties, and microscopic morphologies.
The work is well-written and is of interest to the food industry.
Comments:
- The quality of some figures needs to be improved (i.e. figures 1, 4, 6, and 8).
- What are the chemical compositions (protein, fats, sugars, etc) of different types of milk used in this study? How the differences in the chemical compositions of different kinds of milk can influence the fermentation process.
Author Response
Reviewer 3
The authors investigated the effects of using three types of animal milk (cow, buffalo, and goat) to grow kefir grains from which were used to extract microbial exopolysaccharides. They also characterized the physicochemical properties of the biopolymers in order to establish a preliminary identification, based on their sugar and protein composition, thermal properties, and microscopic morphologies.
The work is well-written and is of interest to the food industry.
Response: We thank the reviewer for his/her positive comments.
Comments:
- The quality of some figures needs to be improved (i.e. figures 1, 4, 6, and 8).
Response: We thank the reviewer for his/her suggestion. We have improved the figures as you suggested.
- What are the chemical compositions (protein, fats, sugars, etc) of different types of milk used in this study? How the differences in the chemical compositions of different kinds of milk can influence the fermentation process.
Response: We thank the reviewer for his/her comment. The chemical composition (g/100 ml of milk) of the three types of milk is indicated below:
UHT cow milk (nutritional values are reported in the nutritional values label of the commercial product):
carbohydrates 5.0 g, fats 3.6 g, proteins 3.4 g and salt 0.15 g
Fresh buffalo milk (nutritional values have been provided by the company)
carbohydrates 5.1 g, fats 8.5 g, proteins 4.5 g and salt 0.2 g
UHT goat milk
carbohydrates 4.7 g, fats 3.5 g, proteins 3.2 g and salt 0.19 g.
In this study, we did not investigate how the different chemical composition of the milks might affect the fermentation process. Indeed the hypothesis of this study is that the different composition of various type of raw milk could lead to different exopolysaccharides. In order to refute or confirm the hypothesis, three types of animal milk (cow, buffalo and goat) were used for the growing of kefir grains from which to extract microbial exopolysaccharides by three methods.
Certainly, the fermentation process is the composite result of the metabolism of a variety of microorganisms which are homofermentative and heterofermentative lactic acid bacteria, lactose-assimilating and non-lactose-assimilating yeasts. During fermentation homofermentative LABs transform lactose into lactic acid and heterofermentative LABs produce formic, lactic and acetic acids, ethanol and carbon dioxide. Excess lactic acid was consumed, along with galactose, by non-lactose-assimilating yeasts as a carbon source and energy source, but some yeasts are proteolytic and lipolytic and they may obtain cell growth components by this metabolism. Yeast an bacteria share a symbiotic relationship meaning that they survive or propagate by sharing their bioproducts as source or growth stimulating source. These complex interactions between bacteria and yeasts that also depend on the composition of the raw milk can influence product characteristics and quality.

Reviewer 4 Report
Interesting work. I have some suggetions for improvment of the manuscript in the attached file.

Author Response
Reviewer 4
Interesting work. I have some suggestions for improvement of the manuscript in the attached file.
Response: We thank the reviewer for his/her useful comment to improve our research article.
We corrected the manuscript as you suggested in the attached file.

Round 2
Reviewer 2 Report
Thank you to the authors for taking into account comments from all the reviewers which already improves the quality of the manuscript.
My last comment concerns Figures 3, 4 and 5 where the attribution of the letters is not appropriate.
Figure 3: using letters "a" from "e" suggested that you compared all conditions together, LAB and Yeast together. I actually think that your statistics are representative of only one class, LAB or Yeat. Therefore, for Yeast, you should use capital letters A, B, C instead of c, d, e.
Similarly for Figures 4 and 5 where statistics seem to be done per milk source only without considering the factorial effect. If that's the case then each milk source (buffalo, cow, goat) should have different statistical symbols. For instance Buffalo : a, b, c, Cow: A, B, C, etc... However, in this case, the statistical analysis should take into account the factorial effect. Indeed, the type of extractions I, II or III, induces different results according to the milk source. Using different letters/symbols per milk sources as mentioned above will already highlight the factorial effect.
Author Response
Thank you to the authors for taking into account comments from all the reviewers which already improves the quality of the manuscript.
My last comment concerns Figures 3, 4 and 5 where the attribution of the letters is not appropriate.
Figure 3: using letters "a" from "e" suggested that you compared all conditions together, LAB and Yeast together. I actually think that your statistics are representative of only one class, LAB or Yeat. Therefore, for Yeast, you should use capital letters A, B, C instead of c, d, e.
Response: We thank the reviewer for his/her suggestion. We are sorry for the inappropriate use of the letters. We have changed the letters as you suggested.
Similarly for Figures 4 and 5 where statistics seem to be done per milk source only without considering the factorial effect. If that's the case then each milk source (buffalo, cow, goat) should have different statistical symbols. For instance Buffalo : a, b, c, Cow: A, B, C, etc... However, in this case, the statistical analysis should take into account the factorial effect. Indeed, the type of extractions I, II or III, induces different results according to the milk source. Using different letters/symbols per milk sources as mentioned above will already highlight the factorial effect.
Response: We thank the reviewer for his/her suggestion. We have changed the letters as you suggested.
Figure 4
Figure 5
The adjusted figures are visible in the uploaded PDF document ( Answer to reviewer 2)
